# OpenReview forum: "When Thinking Drifts: Evidential Grounding for Robust Video Reasoning"
_NeurIPS.cc/2025/Conference — NeurIPS 2025 poster_

### Official Review · Reviewer_1B4M · 2025-07-02

**Clarity:** 3
**Significance:** 2
**Originality:** 2
**Rating:** 4
**Confidence:** 3

**Summary:**

This paper investigates the limitations of Chain-of-Thought (CoT) prompting in video reasoning tasks for multimodal large language models (MLLMs). The authors identify a critical issue termed "visual thinking drift," where CoT-generated reasoning chains diverge from actual visual evidence, leading to hallucinations and degraded performance. They propose a Bayesian-inspired explanation for this phenomenon, attributing it to the amplification of language priors over visual signals during autoregressive decoding. To mitigate this, the authors introduce Visual Evidence Reward (VER), a reinforcement learning framework that explicitly rewards reasoning traces grounded in verifiable visual evidence. Extensive experiments across 10 video understanding benchmarks demonstrate that VER consistently improves accuracy over baseline models.

**Questions:**

1.  How does Video-VER perform on videos exceeding 32 frames? While the ablation study (Table 3) demonstrates improved performance with increased frame counts, real-world applications often involve much longer sequences.
2. How does Video-VER compare to modular reasoning approaches (e.g., [46] or [38]) that explicitly decompose video understanding into subtasks like action recognition or temporal grounding?

**Ethical Concerns:**

["NO or VERY MINOR ethics concerns only"]

**Final Justification:**

The authors have addressed my major concerns. In light of these improvements, I have revised my rating accordingly and now recommend an acceptance.

**Limitations:**

Yes

**Quality:**

3

**Strengths And Weaknesses:**

### Strengths
1. The concept of *visual thinking drift* is an interesting and underexplored challenge in video reasoning. The paper provides empirical findings (Figure 2) and theoretical insights (Bayesian analysis) into such problem.
2. The experiments span 10 diverse benchmarks, demonstrating robustness across tasks like temporal reasoning, object counting, and hallucination mitigation.

### Weaknesses
1. The Bayesian analysis in Section 3.2 attributes *visual thinking drift* to the dominance of language priors over visual evidence during autoregressive decoding. However, this explanation is overly generic and fails to address why CoT prompting works relatively better for text-based LLMs (where "prefill context" acts as the analog of v) compared to video reasoning tasks. This suggests that the root cause of drift may be more nuanced and specific to video reasoning challenges. The authors should explicitly investigate why CoT fails for video tasks even when the same Bayesian mechanism applies to both modalities.
2. The experiments in this paper exclusively use `Qwen2.5-VL-7B` as the base model for evaluating the Video-VER framework. While the method demonstrates improvements on this specific architecture, the lack of testing across diverse model architectures (e.g., LongVA, LLaVA-OneVision) or parameter scales (e.g., smaller or larger models) raises concerns about the generalizability of Video-VER.
3. The generation of visual evidence relies on a strong external MLLM (`Qwen2.5-VL-72B`), which could introduce biases or hallucinations itself. While the paper mentions filtering, the robustness of this process is not fully validated.

---

> ### Author Rebuttal · Authors · 2025-07-31
>
> Thank you for the valuable feedback. We are happy that you appreciated our identification of the **“visual thinking drift” phenomenon as an interesting and underexplored challenge**, as well as our **empirical findings**, the **theoretical insights**, and **the breadth of our experimental evaluation across diverse tasks**.
>
> ---
>
> To address your concerns, we provide the following response:
>
> > **Q1. “explicitly investigate why CoT fails for video tasks even when the same Bayesian mechanism applies to both modalities”**
>
> Thank you for the insightful comment! The “visual thinking drift” phenomenon is indeed specific to video reasoning and deserves clearer explanation.
>
> In L168, we note that language priors outweigh visual likelihoods in practice, but did not elaborate further. In today’s video-LLMs, the language-projection matrix  $W_{lang}$ tends to have a much larger norm than its visual counterpart $W_{vis}$. This is because the model inherits a massive, pre-trained language backbone trained on trillions of text tokens, while the vision pathway is typically a small, newly added adapter, fine-tuned on multimodal data that is orders of magnitude smaller. Moreover, during inference, far fewer visual tokens influence gradient flow compared to language tokens. This asymmetry in initialization, data scale, and gradient dynamics leads to a significantly smaller Frobenius norm in $W_{vis}$, causing language priors to dominate during autoregressive decoding and contributing to the observed drift.
>
> Another important point is the modality mismatch in visual reasoning. In text tasks, the question, answer, and reasoning all happen in the same modality—language—so the model "thinks" in the same space as it reads and answers. But in video tasks, the model must reason about visual content using language as a conduit. This indirect reasoning makes it easier for the model to drift away from the actual video, especially when visual grounding is weak.
>
> We will update this explanation in the text accordingly.
>
> > **Q2. Generalization test across diverse model architectures (e.g., LongVA, LLaVA-OneVision) or parameter scales**
>
> Thank you for the constructive feedback. As noted in [1], LLaVA struggles to generate high-quality, coherent reasoning paths—an observation consistent with our own findings (noted in the footnote under L145). Per the reviewer’s suggestion, we also tested LongVA and observed a similar issue. We conjecture that this limitation stems from the use of a weaker underlying language model (Qwen2), which appears insufficient for reliably supporting structured reasoning prompts.
>
> To our knowledge, Qwen2.5-VL is currently the most stable open-source model for following structured prompts and generating reasoning traces, as supported by recent literature [17, 32] and frequently used open-source projects [2, 3]. This capability is crucial for applying RL-based fine-tuning, which relies on consistent intermediate reasoning.
>
> That said, we appreciate the reviewer’s concern regarding the generalizability of Video-VER. To address this, we conducted additional experiments using the Qwen2.5-VL base model with different parameter scales, 3B. As shown in the table below, our VER method still provides consistent accuracy improvements over the 3B model, demonstrating that our approach generalizes well across model scales.
>
> | Model            | MVBench | Video-MME | VideoMMMU | MMVU  | VideoHal. | EventHal. | VSI-Bench | TempC. | TVBench | Vinog. |
> |------------------|---------|-----------|-----------|-------|----------------|----------------|-----------|--------------|---------|------------|
> | Qwen2.5-VL-3B          | 57.4    | 53.1      | 33.8      | 53.6  | 40.4           | 49.9           | 25.4      | 59.8         | 47.2    | 5.8        |
> | Qwen2.5-VL-3B+VER | 62.7    | 54.5      | 42.3      | 58.2  | 45.6           | 55.7           | 30.3      | 63.4         | 48.2    | 6.8        |
>
> [1] Yang, L., et al. (2025). GRPO-for-Llava. GitHub repository.
>
> [2] Wang, X., et al. (2025). Open-R1-Video. GitHub repository.
>
> [3] Feng, K., et al. (2025). Video-R1. GitHub repository.
>
> > **Q3. Potential hallucination introduced by the 72B visual evidence generator**
>
> We understand that the reviewer's concern lies in the potential biases or hallucinations introduced by the external Qwen2.5-VL-72B as a visual evidence generator. We address this concern in Supplementary Section E. While large models can indeed hallucinate during visual reasoning, our approach is specifically designed to mitigate this issue. We employ several mechanisms to ensure that the generated visual evidence is grounded and visually anchored:
>
> - Offline Use of Teacher Model (supp L381-384): The external 72B MLLM is used only during training to generate question-specific visual evidence. Our Video-VER model learns from this data but operates independently at inference time, ensuring that any potential errors from the teacher model are not propagated or relied upon during deployment.
>
> - Binary Reward Signal (supp L385-391): Instead of requiring exhaustive annotations, we use a binary reward that simply checks whether the generated reasoning includes any verifiable visual fact. If the evidence is missing or incorrect, the model receives a zero reward. This effectively discourages hallucinations without reinforcing incorrect details.
>
> - Inverted Prompting Strategy (supp L392-410): Unlike standard chain-of-thought prompting, we condition the 72B model on both the question and the correct answer, prompting it to produce concise, verifiable visual facts that support the answer. This tightly constrains generation, encouraging relevance and factual grounding over generic or speculative content.
>
> To validate the robustness of our visual evidence extraction method, we provide an ablation study comparing the use and non-use of our Inverted Prompting Strategy (IPS) in generating visual evidence. The results show that IPS significantly improves the quality of visual evidence, which in turn enhances the overall performance of VER across all benchmarks.
>
> In conclusion, through these design choices, our model is trained to prioritize visual grounding over reliance on linguistic priors, directly addressing the root cause of “visual thinking drift.”
>
>
> | Model          | MVBench | Video-MME | VideoMMMU | MMVU  | VideoHal. | EventHal. | VSI-Bench | TempC. | TVBench | Vinog. |
> |----------------|---------|-----------|-----------|-------|----------------|----------------|-----------|--------------|---------|------------|
> | Qwen2.5-VL        | 59.8    | 54.7      | 47.8      | 60.5  | 44.1           | 67.3           | 31.4      | 71.3         | 49.9    | 12.8       |
> | Ours (w/o IPS) | 62.7    | 57.5      | 50.9      | 64.6  | 51.9           | 69.0           | 33.2      | 72.9         | 51.9    | 13.6       |
> | Ours (w/ IPS)  | 64.1    | 59.3      | 52.7      | 65.1  | 53.1           | 70.0           | 34.6      | 74.0         | 52.8    | 14.4       |
>
>
> > **Q4. Performance on videos exceeding 32 frames**
>
> We appreciate the reviewer’s interest in evaluating our method on longer video sequences. While our main experiments (Table 3) focus on input lengths of 8, 16, and 32 frames, we have conducted additional experiments with 64-frame inputs to assess scalability. These results, presented in the table below, show that our method effectively leverages extended temporal context and continues to perform well, demonstrating strong scalability and generalization to longer sequences.
>
> Due to computational constraints, we are limited to evaluating up to 64 frames. However, we agree that handling longer videos is crucial for real-world applications. As highlighted in our failure case analysis (Figure 14), missing key visual cues due to limited sampling or encoding can impair reasoning. Thus, the VER mechanism not only enhances visual reasoning but also stands to benefit from continued advances in long-context video modeling.
> | #Frames   | MVBench | Video-MME | VideoMMMU | MMVU  | VideoHal. | EventHal. | VSI-Bench | TempC. | TVBench | Vinog. |
> |-----------|---------|-----------|-----------|-------|----------------|----------------|-----------|--------------|---------|------------|
> | 32 | 64.1    | 59.3      | 52.7      | 65.1  | 53.1           | 70.0           | 34.6      | 74.0         | 52.8    | 14.4       |
> | 64 | 65.0    | 61.7      | 52.7      | 65.6  | 53.5           | 71.5           | 36.1      | 74.1         | 53.3    | 15.4       |
>
>
> > **Q5. Comparison with [46] or [38]**
>
> As noted in L38–39, both [46] and [38] focus on curating extensive spatio-temporal annotations to support supervised fine-tuning (SFT), where the model learns from human-labeled examples for specific subtasks. In contrast, our work centers on reinforcement fine-tuning (RFT), which directly optimizes visually grounded reasoning through an RL reward signal. This approach enables the model to learn end-to-end reasoning without relying on handcrafted subtask-specific annotations.
>
> We have included additional evaluations of [46] and [38] across four benchmarks. For both baselines, we report their best published results. Our Video-VER model consistently outperforms these methods, suggesting that our visually grounded GRPO framework offers a competitive—and potentially more generalizable—alternative to these modular, annotation-heavy approaches.
>
> | Method            | MVBench | VSI-Bench | NExT-QA | Video-MME |
> |-------------------|---------|-----------|---------|-----------|
> | VITED [38]        | 61.0    | -         | 73.4    | -         |
> | AoTD [46]         | 55.6    | 28.8      | 77.6    | 45.0      |
> | Video-VER (Ours)  | 64.1    | 34.6      | 81.0    | 59.3      |

---

> > ### Comment · Reviewer_1B4M · 2025-08-03
> >
> > Thank you for your detailed responses. You have adequately addressed some of my concerns (Q2, Q4, and Q5). However, I still have two remaining issues that require further clarification or validation:
> >
> > - Regarding Q1: While your explanations are intuitively reasonable, they lack rigorous theoretical or empirical proof.
> > - Regarding Q3: While QA performance indirectly validates visual evidence generation, **direct** assessment (e.g., manual inspection or automated metrics) is needed to confirm its accuracy.

---

> ### Author Response · Authors · 2025-08-05
> **Further clarifications on Q1 and Q3**
>
> We sincerely appreciate the reviewer’s thoughtful comments and are pleased that our responses have satisfactorily addressed the concerns regarding Q2, Q4, and Q5. We would be happy to offer further clarification on Q1 and Q3, as detailed in our separate comments below.

---

> ### Author Response · Authors · 2025-08-05
> **Clarification on Q1 – Part 1**
>
> For Q1, we understand that the reviewer’s primary concern relates to the Bayesian analysis presented in Section 3.2. While our theoretical insights were considered “intuitively reasonable”, the level of detail provided may have appeared moderate, as suggested by the reviewer.
>
> The core question raised is why the same Bayesian mechanism applies to both language and visual modalities, yet the performance degradation introduced by CoT is observed primarily in video-related tasks. In essence, the reviewer is seeking a more nuanced explanation of the root cause of this performance degradation, particularly as it pertains to the unique challenges in video-based tasks. We appreciate this thoughtful question and are happy to provide supplementary justification to further support better understanding of our analysis.
>
> ----
>
> Consider a question $q$, condition $\mathbf v$ (which represents either the video features in video LLMs or the "prefilling text" in text-only LLMs), reasoning tokens $c_{1:T}$ and final answer $a$, the joint probability under an autoregressive video-LLM is
>
> $$
> p(c_{1:T}, a \mid q, \mathbf v)
> \=\
> p\\bigl(a \mid c_{1:T}, q, \mathbf v\bigr)
> \prod_{t=1}^{T} p\\bigl(c_t \mid c_{<t}, q, \mathbf v\bigr).
> \tag{1}
> $$
>
> We now derive an explicit form for each conditional $p\\bigl(c_t \mid c_{<t}, q, \mathbf v\bigr)$.
>
> In video LLMs, at step $t$ the decoder’s last hidden vector can be written as
> $$
> \mathbf h_t
> \=\
> \underbrace{\mathbf h^{(\text{lang})} _t} _{\text{from previous text tokens}}
> \+\
> \underbrace{A _{\text{vis}}\\bigl(\mathbf r_t^{(\text{vis})}\bigr)} _{\text{from visual adapter}},
> \tag{2}
> $$
>
> where $A_{\text{vis}}\in\mathbb R^{d\times d_{\text{vis}}}$ is the vision adapter (either a linear layer or MLP), and $\mathbf r_t^{(\text{vis})}$ is the visual embedding.
>
> During decoding, every token passes through the same vocabulary projection $W_{\text{out}}\in\mathbb R^{d\times|\mathcal V|}$. For a candidate token $w$ with one-hot encoding $\mathbf e_w$, the logit is computed as $z _w=  \mathbf h _t^{\\top} W _{\text{out}} \mathbf e _w$.
>
> Substituting into Equation (2),
>
> $$
> z _w
> \=\
> (\mathbf h^{(\text{lang})} _t)^{\\top} W _{\text{out}}\mathbf e _w
> \+\
> \bigl(A _{\text{vis}}\mathbf r _t^{(\text{vis})}\bigr)^{\\top} W _{\text{out}}\mathbf e _w
> \=\
> \mathbf h _{c _{<t}}^{\\top} \underbrace{W _{\text{out}}} _{W _{\text{lang}}}\mathbf e_w
> \+\
> \mathbf h _{\mathbf v}^{\\top}
> \underbrace{\bigl(A _{\text{vis}}^{\\top} W _{\text{out}}\bigr)} _{W _{\text{vis}}}\\mathbf e _w,
> \tag{3}
> $$
>
> where we define:
> $$
> \mathbf h _{c _{<t}}\equiv \mathbf h^{(\text{lang})} _t,
> \qquad
> \mathbf h _{\mathbf v}\equiv \mathbf r _t^{(\text{vis})},
> \qquad
> W _{\text{vis}} = W _{\text{out}}A _{\text{vis}}.
> $$
>
> The conditional distribution is given by a softmax over all tokens $w$:
> $$
> p(c _t = w \mid c _{\text{<t}}, q, \mathbf v)
> \=\
> \frac{\exp(z _w)}{\displaystyle\sum _{w'\in\mathcal V}\exp(z _{w'})}.
> \tag{4}
> $$
>
> Removing the common denominator yields the unnormalised expression:
> $$
> p\\bigl(c _t \mid c _{\text{<t}}, q, \mathbf v\bigr)
> \\propto\
> \exp\\Bigl(
> \underbrace{\mathbf h _{c _{<t}}^{\\top} W _{\text{lang}}} _{\text{language prior}}
> \+\
> \underbrace{\mathbf h _{\mathbf v}^{\\top} W _{\text{vis}}} _{\text{visual likelihood}}
> \Bigr).
> \tag{5}
> $$
>
> Here $W_{\text{lang}}\equiv W_{\text{out}}$ and $W_{\text{vis}} \equiv W_{\text{out}}A_{\text{vis}}$.
>
> ----
>
> In our previous response, we noted that the dominance of language priors in video-LLMs stems from asymmetries in model initialization and data scale. These asymmetries lead to a substantially smaller visual projection norm, causing language signals to overpower visual inputs during decoding—formally, $\lVert W_{\text{vis}}\rVert_F \ll \lVert W_{\text{lang}}\rVert_F$.
>
> In contrast, for text-only LLMs, there is no visual term $A_{\text{vis}}(\mathbf r _t^{(\text{vis})})$ due to the absence of a vision encoder. As a result, the same "thinking drift" mechanism does not apply in purely textual settings.

---

> ### Author Response · Authors · 2025-08-05
> **Clarification on Q1 – Part 2**
>
> To further substantiate this point, we now provide additional **theoretical** and **empirical evidence** supporting the presence of this asymmetry—specifically, how the dominance of the language component underlies the phenomenon we refer to as *visual thinking drift*.
>
> ----
>
> **Theoretical Justification.** We present a simple closed-form derivation to explain why fewer gradient updates to the vision encoder naturally lead to a smaller Frobenius norm for the visual projection matrix, compared to that of the language model.
>
> Assume a trainable matrix evolving under stochastic gradient descent (SGD) as:
> $$
> W \\gets\ W^{(0)} + \sum_{t=1}^{N} \Delta_t,
> $$
> where the updates $\Delta_t$ have zero mean and second moment
> $\mathbb{E}\bigl[\lVert\Delta_t\rVert_F^{2}\bigr]=\sigma^{2}$.
> Then the expected squared Frobenius norm becomes:
> $$
> \mathbb{E}\bigl[\lVert W\rVert_F^{2}\bigr] \=\\lVert W^{(0)}\rVert_F^{2} + N\sigma^{2},
> \quad\Longrightarrow\quad
> \mathbb{E}\bigl[\lVert W\rVert_F\bigr] \\propto\ \sqrt{N}.
> $$
>
> This shows that the expected Frobenius norm grows with the square root of the number of updates $N$.
>
> In modern video-LLMs, the language pathway typically consists of a massive pre-trained backbone trained on trillions of text tokens (e.g., ~18 trillion for Qwen 2.5). In contrast, the visual pathway often involves a newly introduced adapter module trained on significantly fewer multimodal examples—on the order of tens of billions of paired tokens (e.g., ~40B for Flamingo). As a result, the number of gradient updates $N$ for the visual projection is much smaller, leading naturally to the observed asymmetry: $\lVert W_{\text{vis}}\rVert_F \ll \lVert W_{\text{lang}}\rVert_F$.
>
>
> ----
>
> **Empirical Evidence.** To validate this asymmetry, we inspected two publicly available video-LLM checkpoints—Qwen2.5-VL-7B and LLaVa-OneVision-7B—and measured the Frobenius norms of the visual and language projection matrices. The results are as follows:
>
> | Model            | $\lVert W_{\text{lang}}\rVert_F$ | $\lVert W_{\text{vis}}\rVert_F$ | Ratio |
> |------------------------|---------|-----------|---------|
> | Qwen2.5-VL-7B       |  80.9  | 12.3        | 6.6x    |
> | LLaVa-OneVision-7B         | 68.4    | 17.1      | 4.0x    |
>
> These findings empirically validate the asymmetry discussed above: in both models, the language projection layer has a significantly larger Frobenius norm than its visual counterpart. This suggests that the language signal is considerably stronger during decoding, leading to a dominance of language priors over visual likelihoods [1].
>
> [1] Attention is Not Only a Weight: Analyzing Transformers with Vector Norms, EMNLP 2020.
>
> ----
> We hope this additional theoretical and empirical analysis provides a clearer understanding of how language priors can overpower visual inputs in current video-LLMs, contributing to the phenomenon we term visual thinking drift. Please do not hesitate to let us know if you have any further questions or would like additional clarification.

---

> ### Author Response · Authors · 2025-08-05
> **Clarification on Q3**
>
> For Q3, we thank the reviewer for noting that our additional Video-QA results **already lend indirect support to the pipeline**. Below we provide the requested direct evaluation, combining a controlled **quantitative audit** with **qualitative inspection**.
>
>
> **Quantitative human audit.** We randomly sampled 200 videos from our RL post-training dataset and generated the visual evidence examples with our IPS and without. Three expert annotators independently labelled each sentence as Aligned (fully supported by at least one frame) or Not-Aligned (hallucinated). Ties were adjudicated through majority voting.
>
>
>
> | Model            | Aligned | Not Aligned |
> |------------------------|---------|-----------|
> | Ours w/o IPS	       |  78\%  | 22\%  |
> | Ours w/ IPS         | 92\%   | 8\% |
>
> IPS therefore cuts hallucinations by 14\%, confirming that our approach substantially improves visual-evidence quality. Combined with the binary reward that zeroes out any residual hallucinations during training, this demonstrates the robustness of our evidence-generation process.
>
>
> **Qualitative check.** As shown in Fig. 5 and Fig. 11, our Inverted-Prompting Strategy (IPS) generates visually grounded evidence that correctly references both the relevant event and its temporal context.
>
> ---
>
> Taken together, we think these new audits address the reviewer’s original comment:  “the generation of visual evidence relies on a strong external MLLM (Qwen‑2.5‑VL‑72B), which could introduce biases or hallucinations itself. While the paper mentions filtering, the robustness of this process is not fully validated.”  Concretely, our IPS‑driven filtering offers a 14 % reduction in hallucinations indicating improved robustness.  We appreciate the suggestion; this additional analysis will strengthen the paper.

---

> > ### Comment · Reviewer_1B4M · 2025-08-06
> >
> > Thank you for your detailed responses, which have addressed my concerns. I am willing to raise my rating accordingly.

---

> > > ### Author Response · Authors · 2025-08-06
> > >
> > > Thank you for your thoughtful follow-up and for being open to updating your rating. We're glad to hear that your concerns have been addressed. Please don't hesitate to reach out if any further questions or suggestions arise.

---

### Official Review · Reviewer_9tCR · 2025-07-02

**Clarity:** 3
**Significance:** 3
**Originality:** 2
**Rating:** 4
**Confidence:** 4

**Summary:**

This paper addresses the issue where Chain-of-Thought (CoT) often degrades performance in video reasoning tasks because the model overlooks crucial visual content during the reasoning process. To address this, the paper introduces a new reward mechanism, called Visual Evidence Reward (VER), which penalizes the model when it fails to infer to actual visual details. Specifically, ground truth evidence is generated by prompting the Qwen2.5-VL-72B model with both a video and a question, and an auxiliary LLM compares this ground truth evidence with the policy model’s response to provide a binary signal indicating whether the two are aligned. This VER is integrated into the GRPO framework. Experimental results on various video reasoning benchmarks show improvements over the baseline.

**Questions:**

Unfair Comparison Setup [W1] is the most significant issue affecting my decision. As it stands, I don't believe readers can accurately assess the experimental impact of the proposed method.

Q1. See [W1]

Q2. See [W2]

Q3. **Regarding Evidence Generation as Distillation for VideoQA**: The paper uses a significantly larger model (Qwen2.5-VL-72B) to generate the visual evidence that guides the training of Video-VER-7B. Could the authors discuss this process from the perspective of knowledge distillation – where the 7B model is learning to better ground its VideoQA reasoning by aligning with the evidence identified by the 72B model? Is the 72B model expected to perform clearly better overall due to its size, or does the VER training enable the smaller 7B model to achieve comparable or even superior performance in VideoQA reasoning on the target benchmarks?

**Ethical Concerns:**

["NO or VERY MINOR ethics concerns only"]

**Final Justification:**

My major initial concern was unfair comparison setup, and this was well addressed in the authors' rebuttal. Comparing individual gains indeed have shown that VER can present unique gains compared to naive GRPO training. Other remaining concerns were about whether the proposed method ties the learning to the teacher model or can learn beyond their visual evidence. This was also well addressed in the rebuttal phase. Therefore, with these major concerns resolved, I am inclined to revise my rating to 4: borderline accept.

**Limitations:**

yes

**Quality:**

3

**Strengths And Weaknesses:**

## Strengths

S1. **Interesting and Sound Motivation**: Section 3 presents interesting and meaningful take-away to readers that CoT does not exactly help VideoLLMs to perform temporal reasoning because it actually ignores videos.

S2. **Clear Presentation:** The paper is well-written and easy to follow, with clear explanations that make the context easy to understand.

---
## Weaknesses

While this paper tackles an interesting problem, I have major concerns regarding the method and experimental validations:

W1. **Unfair Comparison Setup:** To provide a fair evaluation, there should be an ablation study comparing the following setups: baseline+SFT, baseline+SFT+GRPO, and baseline+SFT+GRPO+VER. In the current experimental setup, it is unclear whether the observed gains are due to GRPO or VER, making it difficult for readers to assess the individual contributions.

W2. **Reliability of Visual Evidence Source**: Figure 4 indicates that even strong reasoning models, such as GPT-4o, can disregard visual facts. However, the paper then uses Qwen2.5-VL-72B to generate the visual evidence that grounds the training. This raises natural questions about how Qwen2.5-VL-72B is free from the same issue of potentially overlooking or hallucinating visual information when generating this critical evidence.

---

> ### Author Rebuttal · Authors · 2025-07-31
>
> Thank you for appreciating the **interesting and well-founded insight into the “visual thinking drift” phenomenon** presented in our work, as well as our overall **clear presentation**.
>
> ---
>
> We understand that the reviewer's main concern lies in the *less well-explained cause of the observed accuracy improvements* in our experimental results. To address this, we provide the following response.
>
> > **Q1. Comparison Setup: Is the gain from GRPO or VER? – “most significant issue affecting my decision”**
>
> Thanks for the thoughtful feedback! We agree that providing a clearer breakdown of the contributions from each model component is important for evaluating the effectiveness of our approach. In response, we present detailed ablation results below that compare the requested configurations: Baseline + SFT, Baseline + SFT + GRPO, and Baseline + SFT + GRPO + VER (full model). Baseline here is Qwen2.5-VL-7B model.
>
> Our findings show that fine-tuning the baseline model with SFT alone does not consistently improve performance. Adding GRPO yields moderate gains across most benchmarks. Notably, incorporating our Visual Evidence Reward (VER) on top of GRPO leads to substantial additional improvements. All performance deltas are reported relative to the original baseline model.
>
> These results help isolate the contribution of each component and demonstrate that the gains observed in our full model are primarily driven by VER, validating its effectiveness in promoting grounded and accurate reasoning.
>
> | Method                   | MVBench     | Video-MME   | VideoMMMU   | MMVU        | VideoHallucer |
> |--------------------------|-------------|-------------|-------------|-------------|----------------|
> | Baseline                     | 59.8        | 54.7        | 47.8        | 60.5        | 44.1           |
> | Baseline + SFT              | 60.5 (+0.7) | 55.4 (+0.7) | 48.2 (+0.4) | 63.5 (+3.0) | 46.9 (+2.8)    |
> | Baseline + SFT + GRPO       | 61.2 (+1.4) | 54.8 (+0.1) | 49.6 (+1.8) | 63.4 (+2.9) | 47.8 (+3.7)    |
> | Baseline + SFT + GRPO + VER | 64.1 (+4.3) | 59.3 (+4.6) | 52.7 (+4.9) | 65.1 (+4.6) | 53.1 (+9.0)    |
>
> | Method                   | EventHallusion | VSI-Bench   | TempCompass | TVBench     | Vinoground  |
> |--------------------------|----------------|-------------|--------------|-------------|-------------|
> | Baseline                     | 67.3           | 31.4        | 71.3         | 49.9        | 12.8        |
> | Baseline + SFT              | 65.5 (-1.8)    | 32.7 (+1.3) | 68.8 (-2.5)  | 50.0 (+0.1) | 10.4 (-2.4) |
> | Baseline + SFT + GRPO       | 67.5 (+0.2)    | 33.2 (+1.8) | 69.7 (-1.6)  | 51.3 (+1.4) | 12.6 (-0.2) |
> | Baseline + SFT + GRPO + VER | 70.0 (+2.7)    | 34.6 (+3.2) | 74.0 (+2.7)  | 52.8 (+2.9) | 14.4 (+1.6) |
>
>
> > **Q2. Reliability of Visual Evidence Source**
>
> We understand that the reviewer's concern lies in the reliability of Qwen2.5-VL-72B for generating visual evidence, particularly given its potential to overlook or hallucinate visual details—an issue also observed in models like GPT-4o. We address this concern in Supplementary Section E. While large models can indeed hallucinate during visual reasoning, our approach is specifically designed to mitigate this issue. We employ several mechanisms to ensure that the generated visual evidence is grounded and visually anchored:
>
> - Offline Use of Teacher Model (supp L381-384): The external 72B MLLM is used only during training to generate question-specific visual evidence. Our Video-VER model learns from this data but operates independently at inference time, ensuring that any potential errors from the teacher model are not propagated or relied upon during deployment.
>
> - Binary Reward Signal (supp L385-391): Instead of requiring exhaustive annotations, we use a binary reward that simply checks whether the generated reasoning includes any verifiable visual fact. If the evidence is missing or incorrect, the model receives a zero reward. This effectively discourages hallucinations without reinforcing incorrect details.
>
> - Inverted Prompting Strategy (supp L392-410): Unlike standard chain-of-thought prompting, we condition the 72B model on both the question and the correct answer, prompting it to produce concise, verifiable visual facts that support the answer. This tightly constrains generation, encouraging relevance and factual grounding over generic or speculative content.
>
> To demonstrate the effectiveness of our visual evidence extraction method, we provide an ablation study comparing the use and non-use of our Inverted Prompting Strategy (IPS) in generating visual evidence. The results show that IPS significantly improves the quality of visual evidence, which in turn enhances the overall performance of VER across all benchmarks.
> In conclusion, through these design choices, our model is trained to prioritize visual grounding over reliance on linguistic priors, directly addressing the root cause of “visual thinking drift.”
>
> | Model          | MVBench | Video-MME | VideoMMMU | MMVU  | VideoHal. | EventHal. | VSI-Bench | TempC. | TVBench | Vinog. |
> |----------------|---------|-----------|-----------|-------|----------------|----------------|-----------|--------------|---------|------------|
> | Qwen2.5-VL        | 59.8    | 54.7      | 47.8      | 60.5  | 44.1           | 67.3           | 31.4      | 71.3         | 49.9    | 12.8       |
> | Ours (w/o IPS) | 62.7    | 57.5      | 50.9      | 64.6  | 51.9           | 69.0           | 33.2      | 72.9         | 51.9    | 13.6       |
> | Ours (w/ IPS)  | 64.1    | 59.3      | 52.7      | 65.1  | 53.1           | 70.0           | 34.6      | 74.0         | 52.8    | 14.4       |
>
>
>
> > **Q3. Regarding Evidence Generation as Distillation for VideoQA**
>
> Thank you for the insightful question! Distillation is an interesting interpretation, and indeed the larger 72B model aims to ensure the most reliable visual evidence signal as possible.  Nonetheless, our method goes well beyond standard knowledge distillation paradigms. Specifically, our model is not trained to mimic the final outputs or answer distributions of the 72B model through supervised fine-tuning (SFT). Instead, the 72B model is used once, offline, to generate minimal, question-specific visual evidence that supports correct answers. This evidence is then used to define a binary, fact-level reward signal that guides the training of the 7B Video-VER model via reinforcement learning.
>
> This setup differs fundamentally from conventional knowledge distillation in several key ways. **First**, there is no direct imitation of the teacher’s answers. The 7B model is not learning to reproduce the 72B model’s outputs or full reasoning traces. It is rewarded only when its own reasoning includes verifiable visual facts aligned with the evidence. **Second**, the supervision is task-specific and grounded. The visual evidence highlights what matters visually in the context of each question, teaching the model how to reason rather than simply what to predict. **Third**, the 72B model’s role is limited to only helping the smaller model identify and attend to relevant visual cues. There is nothing more.
>
> As a result, the 7B Video-VER model learns to ground its answers in concrete visual evidence, enhancing both interpretability and robustness. In short, our method uses the 72B model as a generator of visual signals that shape the reward function, enabling the smaller model to learn how to reason visually. This distinction is central to the effectiveness of our VER training approach.

---

> ### Comment · Reviewer_9tCR · 2025-08-04
>
> I thank the authors for their efforts in providing detailed responses and additional experiments. The added experiments in the answer to Q1 comparing individual gains indeed show that VER can present unique gains compared to naive GRPO training. The answers to Q2 and Q3 also directly addressed my concerns regarding whether this method ties the learning to the teacher model (72B model) or can learn beyond their visual evidence. While I also have to discuss with other reviewers, I intend to raise my score.

---

> > ### Author Response · Authors · 2025-08-05
> > **Thank You for the Thoughtful Review and Encouraging Feedback**
> >
> > We sincerely thank Reviewer 9tCR for the thoughtful feedback and for taking the time to carefully consider our responses and additional experiments. We’re glad that our clarifications and results helped address your concerns—particularly regarding the unique gains of VER compared to naive GRPO training, and the capacity of our method to learn beyond the visual evidence provided by the teacher model.
> >
> > We’re also grateful to hear that you intend to raise your score, and we deeply appreciate your constructive engagement throughout the review process.

---

### Official Review · Reviewer_Z3Pb · 2025-07-07

**Clarity:** 3
**Significance:** 3
**Originality:** 2
**Rating:** 5
**Confidence:** 3

**Summary:**

This paper presents a systematic analysis revealing that CoT often degrades performance in video reasoning, which is related to a phenomenon called "visual thinking drift." Then, the authors introduce a reinforcement framework, “Visual Evidence Reward (VER)”, to explicitly rewards the generation of reasoning traces with visual evidences. 10 diverse video understanding benchmarks demonstrates the Video-VER achieves top performance compared to other baseline models. This work pointed out the importance of visual grounding in the thinking process of multimodal models, and may encourage the development of large models in robust visual grounding-based rational thinking process.

**Questions:**

- If visual evidence reward can help improve model performance, which performs better between using agentic workflow and end-to-end training? And what is the foundational reason to lead to the difference?
- If the large model can control the behavior in visual space, will there also be hallucinations? Is it consistent with the cause of visual perception illusion?

**Ethical Concerns:**

["NO or VERY MINOR ethics concerns only"]

**Final Justification:**

The author's response resolved most of the issues and the scores were appropriately adjusted.

**Limitations:**

yes

**Paper Formatting Concerns:**

nothing

**Quality:**

3

**Strengths And Weaknesses:**

Strengths：
- Present a phenomenon called visual thinking drift, which is consistent with the well-known AI hallucination problem. This paper further analyze this problem from the perspective of visual grounding, and then propose corresponding RL solutions.
- Conduct extensive experiments to show that Video-VER achieves better performance. Also, the ablation study provides more useful information to show the effects on different models types and model sizes.

Weakness:
- Maybe more reasoning models should be added as baselines, such as OpenAI o1/o3.

---

> ### Author Rebuttal · Authors · 2025-07-31
>
> Thank you for the valuable feedback. We appreciate the acknowledgment of our **analysis of the visual thinking drift phenomenon**, the **further exploration from a visual grounding perspective**, and the **proposed reinforcement learning solutions**, as well as the **extensive experiments** and **ablation study**.
>
> ---
>
> To respond to your questions:
>
> > **Q1: “Maybe more reasoning models should be added as baselines, such as OpenAI o1/o3.”**
>
> Thank you for the suggestion. As shown in Table 1, our baselines span the full spectrum of video-LLM capabilities: proprietary (GPT-4o) and open-source; generalist (LongVA, Video-UTR, LLaVA-OneVision, Kangaroo, Qwen2.5-VL) and specialist video-reasoning models (TinyLLaVA-Video-R1 and Video-R1). Nonetheless, we appreciate the reviewer’s interest in advanced proprietary reasoning models such as OpenAI’s o1/o3. To address this, we evaluated o3 (released April 2025) under both direct-answer (DA) and chain-of-thought (CoT) prompting across all ten video benchmarks. As the table below shows, CoT prompting induces a consistent performance drop—despite o3’s strong reasoning capabilities—**mirroring the trends in Figure 2 and reinforcing our conclusion that CoT does not reliably boost video-LLM performance**.
> | Prompt | MVBench | Video-MME | VideoMMMU | MMVU  | VideoHal. | EventHal. | VSI-Bench | TempC. | TVBench | Vinog. |
> |--------|---------|-----------|-----------|-------|----------------|----------------|-----------|--------------|---------|------------|
> | DA     | 69.7    | 74.1      | 77.2      | 80.5  | 66.2           | 82.9           | 49.3      | 83.4         | 65.9    | 52.8       |
> | COT    | 66.4    | 71.0      | 71.0      | 78.9  | 63.8           | 82.1           | 37.7      | 81.3         | 62.1    | 36.6       |
>
> (Note: OpenAI’s “Hiding the Chains of Thought” policy prevents o3 from exposing its internal reasoning; in some CoT runs the model interjects messages like “Sorry, I can’t share my private reasoning,” which may contribute to the accuracy loss.)
>
> > **Q2: Which performs better between using agentic workflow and end-to-end training?**
>
> The performance of agentic workflow approaches is often constrained by the capabilities of their component visual specialist models (e.g., key-frame selectors or object detectors). In contrast, end-to-end training leverages a single multimodal LLM trained on large-scale vision-language data, with the vision encoder handling perception and the language model handling reasoning—enabling a more unified approach to video understanding. Our VER model follows this end-to-end paradigm.
>
> As a reference, we include additional evaluations of recent agentic workflow methods [64, 53, 41], which approach video understanding by decomposing the task into sub-problems—such as key-frame selection or temporal grounding—and addressing each with specialized models. Across benchmarks, our 7B Video-VER model demonstrates superior overall performance compared to these baselines.
>
> |   Category     |         Model           | NExT-QA | EgoSchema |
> |-----------------------------|---------|-----------|------------------|
> | Agentic Workflow | SeViLA [64], NeurIPS 2023   | 73.8    | -         |
> | Agentic Workflow | VideoAgent [53], ECCV 2024  | 71.3    | 54.1      |
> | Agentic Workflow | MoReVQA [41], CVPR 2024     | 69.2    | 51.7      |
> | End-to-End Training         |   Video-VER (Ours)      | 81.0    | 66.2      |
>
> Nonetheless, agentic workflows offer some flexibility at test time. Their ability to decompose tasks and incorporate external visual models may provide advantages in specific scenarios.
>
> > **Q3: If the large model can control the behavior in visual space, will there also be hallucinations? Is it consistent with the cause of visual perception illusion?**
>
> Thank you for the thoughtful question. If a large model could directly manage visual information—for example, by maintaining spatially accurate representations or consistently referring to visual evidence throughout the reasoning process—it would likely help reduce hallucinations. However, hallucinations can still arise due to inherent challenges in video perception, such as low visual resolution, ambiguous scenes, or limited temporal grounding.
>
> As discussed in Section 3.2, current video language models (video-LLMs) often rely heavily on language-based priors because visual signals tend to fade or weaken during the model’s step-by-step reasoning process (autoregressive decoding). As a result, the model may generate outputs that are only loosely grounded in the actual video content.
>
> This is where our approach becomes *especially important*. A key focus of our work is to ensure that the model's reasoning actively refers back to visual evidence at intermediate steps. By reinforcing this connection between reasoning and visual input, we aim to reduce hallucinations and improve the model’s alignment with the video. This aligns with our hypothesis that "visual thinking drift" occurs when reasoning becomes unanchored from the visual content, leading the model to fabricate visual details due to insufficient fidelity and control over the visual stream.

---

> > ### Comment · Reviewer_Z3Pb · 2025-08-06
> >
> > Thank you for your responses, which have addressed  my concerns. I will adjust the score accordingly.

---

> > > ### Author Response · Authors · 2025-08-06
> > >
> > > Thank you, Reviewer Z3Pb, for your thoughtful review and for taking the time to reconsider your score. We're glad to hear that your concerns have been addressed. Please feel free to reach out if you have any further questions or feedback.

---

### Official Review · Reviewer_JHn6 · 2025-07-07

**Clarity:** 4
**Significance:** 3
**Originality:** 3
**Rating:** 5
**Confidence:** 4

**Summary:**

"Visual thinking drift" is the phenom. where CoT generates misleading internal descriptions and steps which hinder performance in visual understanding tasks. The authors propose a "visual evidence reward" RL framework to overcome this.

First they motivate this drift problem with an analysis of the success rate of 4 models on 20 video QA tasks with and without using CoT. In many cases, using CoT causes a performance **drop**, demonstrating that visual thinking drift is a real issue. Even with GPT4o, it's better to adopt the direct answer strategy over the CoT strategy for 50% of the video QA benchmarks, contra the strict gain under CoT we observe in the text domain.

Their analysis centers the idea that a "latent state" of each step of a reasoning chain can drift away from the semantics of the vision tokens as the influence of the vision tokens is overwhelmed by influence of prior latent states.

Visual evidence reward is based on this idea: each reasoning step is checked against the visual info using an LM judge to train a policy model. This VER is added to the GRPO reward during RL. Specifically, this judging is performed by comparing the text of the step to Qwen2.5-VL-72B-generated descriptions of the visual input conditioned on the known question for the training example. They train the model using Video-R1-COT-165k for SFT followed by GRPO with VER on Video-R1-260k and Reversed-in-Time.

Their model achieves significant gains over open baselines, particularly in the COT setting.

(Deleted the first review post because I accidentally submitted a review for a different paper)

**Questions:**

Revisiting the demonstrations in Figures 2, 3, and 4 which motivate the problem of visual thinking drift, can you report numbers for Video-VER *without* chain-of-thought? If your hypothesis is true, I would expect to see more consistent performance gains with CoT.

Nitpick: Consider revising to remove  LM-generated flowerly language, such as: "As the chef **deftly** chops vegetables...**envisioning the delicious outcome**." (bolded) Such superfluous adjectives and editorializing are distracting

Beyond this I found the paper quite easy to understand and have no other significant questions or complaints

**Ethical Concerns:**

["NO or VERY MINOR ethics concerns only"]

**Final Justification:**

The authors have slightly increased the quality of the results by adding an ablation on CoT. I think the paper remains a clear Accept.

**Limitations:**

Yes

**Quality:**

4

**Strengths And Weaknesses:**

Solid demonstration of the initial phenomenon (visual thinking drift) which their intervention is intended to fix, with analysis of existing models.

Simple but effective fix to this problem using VER.

Comprehensive set of baselines evaluated.

I see no glaring reasons to reject.

---

> ### Author Rebuttal · Authors · 2025-07-31
>
> Thank you very much for your valuable comments. We are pleased that you appreciated our **“solid demonstration of the initial phenomenon”**, **“analysis of existing models”**, **“simple but effective fix to this problem”**, and **“comprehensive set of baselines evaluated”**.
>
> ---
>
> > **Q1. Results of Video-VER without *chain-of-thought*.**
>
> As shown in the table below, we evaluated our Video-VER model using two types of prompts: a chain-of-thought (CoT) prompt, which encourages the model to reason step by step before answering, and a direct answer (DA) prompt, which requires only a final answer without intermediate reasoning. Both the base model Qwen2.5-VL and our Video-VER model have 7B parameters.
>
> Overall, as the reviewer anticipated, Video-VER achieves more consistent performance improvements when prompted with CoT, which aligns with its training objective. While DA prompting also yields performance gains over the base model on most benchmarks, the improvements are less consistent and generally smaller than those observed with CoT. These results support our hypothesis that VER rewards in GRPO training encourage grounded, step-by-step reasoning.
>
>
> | Model      | Prompt | MVBench     | Video-MME   | VideoMMMU   | MMVU        | VideoHallucer   |
> |------------|--------|-------------|-------------|-------------|-------------|-------------|
> | Qwen2.5-VL | DA     | 63.6        | 59.2        | 47.3        | 64.2        | 51.8        |
> | Qwen2.5-VL | COT    | 59.8        | 54.7        | 47.8        | 60.5        | 44.1        |
> | Video-VER  | DA     | 63.6 (+3.8) | 57.5 (+2.8) | 48.6 (+0.8) | 64.5 (+4.0) | 51.9 (+7.8) |
> | Video-VER  | COT    | 64.1 (+4.3) | 59.3 (+4.6) | 52.7 (+4.9) | 65.1 (+4.6) | 53.1 (+9.0) |
>
> | Model      | Prompt | EventHallusion   | VSI-Bench   | TempCompass      | TVBench     | Vinoground      |
> |------------|--------|-------------|-------------|-------------|-------------|-------------|
> | Qwen2.5-VL | DA     | 64.5        | 32.3        | 73.7        | 52.2        | 14.8        |
> | Qwen2.5-VL | COT    | 67.3        | 31.4        | 71.3        | 49.9        | 12.8        |
> | Video-VER  | DA     | 65.5 (-1.8) | 32.7 (+1.3) | 72.9 (+1.6) | 52.6 (+2.7) | 13.8 (+1.0) |
> | Video-VER  | COT    | 70.0 (+2.7) | 34.6 (+3.2) | 74.0 (+2.7) | 52.8 (+2.9) | 14.4 (+1.6) |
>
> > **Q2. Writing nitpick.**
>
> Thank you for the insightful suggestion! We will revise the text to remove superfluous adjectives.

---

> > ### Comment · Reviewer_JHn6 · 2025-08-09
> >
> > Thank you for adding the results! It gives us a clearer picture of the benefits of your technique that the CoT-based prompt does continue to augment Video-VER and the advantage isn't purely down to just the strength of the model. I remain inclined to clearly Accept and encourage the other reviewers to do the same.

---

### Note · Authors · 2025-08-14

We thank the AC and reviewers for their constructive engagement during the discussion phase. We are happy that we were able to provide additional experiments, clarifications, and analysis in response to the points raised. All four reviewers expressed appreciation for the responses, with three indicating they intend to raise scores.

- **Reviewer-JHn6 (“*inclined to clearly Accept, encourage the other reviewers to do the same*”)**: Praised the “solid demonstration of the initial phenomenon” and “simple but effective fix to this problem using VER,” the “analysis of existing models,” and the “comprehensive set of baselines evaluated.” Also noted the clarity and ease of understanding of the paper.

- **Reviewer-Z3Pb (borderline accept, “*will adjust the score accordingly*”)**: Described the phenomenon and our solution as “important” and likely to “encourage the development of large models in robust visual grounding-based rational thinking process.” Also commended the connection to the well-known hallucination problem, the further analysis from the perspective of visual grounding, the extensive experiments, and the informative ablation study across different model types and sizes.

- **Reviewer-9tCR (“*intend to raise my score*”)**: Praised the “interesting and sound motivation” and “clear presentation.” The request for clearer separation of VER and GRPO contributions was addressed with detailed ablations showing gains attributable to VER. Also noted the additional clarifications on the reliability of visual evidence.

- **Reviewer-1B4M (“*willing to raise my rating accordingly*”)**: Highlighted the “interesting and underexplored challenge” of visual thinking drift and the “empirical findings and theoretical insights” provided, along with the breadth of experiments spanning 10 diverse benchmarks. The questions on the theoretical explanation and robustness of visual evidence were followed by our extended analysis, empirical projection-norm inspection, and a direct human audit.

We are grateful for the opportunity to strengthen the paper through this process, and we thank everyone again for their time and thoughtful consideration of our work.

---

### Decision · Program_Chairs · 2025-09-17

**Decision:**

Accept (poster)

**Comment:**

This paper studies visual reasoning in multimodal large language models (MLLMs). The key problem and motivation is that Chain-of-Thought (CoT) mechanism in video understanding remains underexplored. This work found a visual thinking drift phenomenon, CoT often degrading performance in video reasoning and leading to hallucinated visual details, and further explained the visual thinking drift via a Bayesian lens. A reinforcement learning framework Visual Evidence Reward (VER) is further proposed to explicitly reward reasoning steps aligned with verifiable visual evidence.

The main strengths are that (1) the visual thinking drift problem is novel, (2) the theoretical and empirical analyses are insightful, (3) the proposed visual evidence reward is simple but effective, (4) the evaluation on 10 benchmarks is comprehensive. The main weaknesses or concerns were that (1) comparison with more reasoning models are to be included, (2) a clearer ablation setup to evaluate VER, (3) reliability conresponding to Qwen2.5-VL-72B usage. After rebuttal, the concerns have been addressed and all the final ratings are positive. AC agree with reviewers and recommend accepting this paper.